# eva: Evaluation framework for pathology foundation models

**kaiko.ai**[*]                                                                EVA@KAIKO.AI
**Ioannis Gatopoulos**                                                 IOANNIS@KAIKO.AI
**Nicolas Känzig**                                                     NICOLAS@KAIKO.AI
**Roman Moser**                                                          ROMAN@KAIKO.AI
**Sebastian Otálora**                                                SEBASTIAN@KAIKO.AI

**Editors:** Accepted for publication at MIDL 2024

## Abstract

In computational pathology, self-supervised trained foundation models (FM) surpass supervised ones in scale and performance. However, the benchmarking of FMs remains a challenge due to the diversity in tasks and evaluation methods. To address this, we introduce eva[1], an open-source framework for evaluating computational pathology FMs. eva is designed to be modular and adaptable to both off-the-shelf and customized datasets, metrics, evaluation protocols and model architectures. We benchmark leading pathology FMs across diverse downstream classification tasks, establishing the first public reproducible pathology FM leaderboard and advocating for standardized FM evaluation practices.

**Keywords:** evaluation framework, foundation models, pathology, oncology

## 1. Introduction

Computational pathology, leveraging whole slide images (WSI), holds significant promise for advancing medical diagnostics and disease understanding (Song et al., 2023; Raciti et al., 2023). Yet, the cost of acquiring labeled WSIs for training supervised models, typically limited to specific tasks, highlights the need for more versatile approaches (Guan and Liu, 2021). Foundation models (FMs), trained on large unlabeled datasets, emerge as a viable solution. The embeddings produced by FMs exhibit strong generalization capabilities, enabling them to perform well across a range of *downstream tasks* (Caron et al., 2021; Oquab et al., 2024). However, their non-interpretable nature poses challenges in domain-specific applicability, often leading to unclear and non-reproducible evaluation and benchmarking practices. Despite the availability of public benchmark datasets (Veeling et al., 2018; Kather et al., 2019; Aresta et al., 2019; Wei et al., 2021), standardization of metrics (Reinke et al., 2022), and evaluations (Laleh et al., 2021; Chen et al., 2024; Vorontsov et al., 2024; Kang et al., 2023), there remains a significant gap: the absence of a cohesive, open-source framework that integrates these elements into a unified, reproducible evaluation process.

To this end, we introduce eva: an open-source framework for standardized, reproducible and fair FM-evaluation across diverse pathology tasks. eva has built-in support for numerous publicly available computational pathology datasets and models and is adaptable to customization. Through this work, we show how eva seamlessly facilitates consistent pathology FM evaluation, resulting in reliable outcomes regardless of model size or architecture. This effort contributes to the development of a reproducible and transparent public model leaderboard of pathology FMs. Ongoing work includes incorporating oncology-related tasks for deeper insights into FM capabilities.

---

[*] All authors contributed equally. Names are ordered alphabetically.

1. eva is released under the Apache 2.0 license and is available at https://kaiko-ai.github.io/eva.

## 2. Setup

### 2.1. Linear evaluation protocol

To evaluate the learned visual representations of FMs on patch-level datasets, we follow the widely used linear evaluation protocol (Kolesnikov et al., 2019; Chen et al., 2020; Caron et al., 2021; Vorontsov et al., 2024), where a linear classifier is trained on the embeddings of a frozen FM backbone, and the validation/test accuracy is used as a proxy for representation quality. Through this method, we aim to determine if the embedding space rendered by the FMs captures enough information to solve diverse downstream tasks.

Table 1: Linear evaluation protocol.

| Data transforms | Scale and Crop |
| --- | --- |
| Backbone | frozen |
| Hidden layers | None |
| Dropout | 0.0 |
| Activation function | None |
| Number of steps | 12500 |
| Batch size | 4096 |
| Learning rate | 0.01 |
| End learning rate | 0.0 |
| Early stopping | [Number of steps] * 5% |
| Optimizer | SGD |
| Momentum | 0.9 |
| Weight Decay | 0.0 |
| Nesterov momentum | True |
| LR Schedule | Cosine without warmup |

For consistency with prior literature and fair evaluation, we specify a set of simple and robust default parameters to fit the projection head, avoiding bias towards specific FM backbone architectures. In particular, we follow a configuration where an initial low learning rate gradually diminishes to zero across numerous training iterations to ensure convergence (Chen et al., 2020; Caron et al., 2021; Vorontsov et al., 2024). For smaller datasets, where the proposed batch size is larger than the training dataset (e.g. BACH), we reduce the batch size and linearly scale the learning rate accordingly. For further details about the configuration, refer to Table 1.

While eva provides a broad range of standard metrics, in this article we report *balanced accuracy* throughout the provided benchmarks. This choice aims to prevent number overflow and improve readability, while ensuring a fair representation of model performance, particularly for class-imbalanced datasets. Additionally, we report the average over five independent fitting runs using different seeds along with their standard deviation.

### 2.2. Datasets

We employ four widely-used patch-level classification benchmarks that encompass varying numbers of samples, magnifications, and tissue types, providing valuable insight into the generalizability and overall performance of a FM. A summary of their distinct characteristics is outlined in Table 2.

Consistent with common benchmark practices for self-supervised models evaluation (He et al., 2019; Caron et al., 2021), the linear head is trained on the embeddings of the training set, evaluated on the validation, and where applicable, on the test (e.g. *PCam*). All image patches undergo an identical sequence of transformations: the larger image dimension is scaled to 224 before being center cropped to a 224×224 patch, ensuring the original aspect ratio is maintained without distortion. Finally, the pixel values are normalized with the same normalization constants applied during training.

Table 2: Summary of patch-level benchmarks classification datasets.

| Dataset | # patches | size | magnification (µm/px) | classes | tissue type |
|---|---|---|---|---|---|
| BACH (Aresta et al., 2019) | 400 | 2048×1536 | 20× (0.42 µm/px) | 4 | Breast |
| CRC (Kather et al., 2019) | 107,180 | 224×224 | 20× (0.50 µm/px) | 9 | Colorectal |
| MHIST (Wei et al., 2021) | 3,152 | 224×224 | 5× (2.00 µm/px)$^2$ | 2 | Colorectal |
| PCam (Veeling et al., 2018) | 327,680 | 96×96 | 10× (0.97 µm/px)$^2$ | 2 | Breast |

## 3. Leaderboard

We utilized `eva` to benchmark a set of open-source models on patch-level pathology tasks. The resulting scores are presented in Table 3. Notably, pathology image pre-trained FMs (below the dashed line) consistently outperformed those based on common images (above the dashed line) across all datasets. The leaderboard shows that there is no consistent winner across all benchmark datasets, emphasizing the importance of measuring performance over a diverse set of downstream tasks when developing FMs. Finally, the consistently low standard deviation values indicate that the linear heads converged under the defined configuration, validating the suitability of the linear protocol for evaluation purposes.

Table 3: Linear probing evaluation of FMs, averaged *balanced accuracy* (and standard deviation) over five runs with different random initializations for each dataset.

| Model | BACH | CRC | MHIST | PCam/val | PCam/test |
|---|---|---|---|---|---|
| ViT-S16 *(random init weights)* | 0.410 (±0.009) | 0.617 (±0.008) | 0.501 (±0.004) | 0.753 (±0.002) | 0.728 (±0.003) |
| DINO ViT-S16 (Caron et al., 2021) | 0.695 (±0.004) | 0.935 (±0.003) | 0.831 (±0.002) | 0.864 (±0.007) | 0.849 (±0.007) |
| DINO ViT-B8 (Caron et al., 2021) | 0.710 (±0.007) | 0.939 (±0.001) | 0.814 (±0.003) | 0.870 (±0.003) | 0.856 (±0.004) |
| DINOv2 ViT-L14 (Oquab et al., 2024) | 0.707 (±0.008) | 0.916 (±0.002) | 0.832 (±0.003) | 0.873 (±0.001) | 0.888 (±0.001) |
| DINO$_{(p=16)}$ (Kang et al., 2023) | 0.801 (±0.005) | 0.934 (±0.001) | 0.768 (±0.004) | 0.889 (±0.002) | 0.895 (±0.006) |
| Phikon (Filiot et al., 2023) | 0.725 (±0.004) | 0.935 (±0.001) | 0.777 (±0.005) | 0.912 (±0.002) | 0.915 (±0.003) |
| UNI (Chen et al., 2024) | 0.814 (±0.008) | 0.950 (±0.001) | 0.837 (±0.001) | 0.936 (±0.001) | 0.938 (±0.001) |
| DINO ViT-S16 (kaiko.ai et al., 2024) | 0.797 (±0.003) | 0.943 (±0.001) | 0.828 (±0.003) | 0.903 (±0.001) | 0.893 (±0.005) |
| DINO ViT-S8 (kaiko.ai et al., 2024) | 0.834 (±0.012) | 0.946 (±0.002) | 0.832 (±0.006) | 0.897 (±0.001) | 0.887 (±0.002) |
| DINO ViT-B16 (kaiko.ai et al., 2024) | 0.810 (±0.008) | 0.960 (±0.001) | 0.826 (±0.003) | 0.900 (±0.002) | 0.898 (±0.003) |
| DINO ViT-B8 (kaiko.ai et al., 2024) | 0.865 (±0.019) | 0.956 (±0.001) | 0.809 (±0.021) | 0.913 (±0.001) | 0.921 (±0.002) |
| DINOv2 ViT-L14 (kaiko.ai et al., 2024) | 0.870 (±0.005) | 0.930 (±0.001) | 0.809 (±0.001) | 0.908 (±0.001) | 0.898 (±0.002) |

## 4. Conclusion & Future Work

We introduced `eva`, a versatile evaluation framework designed for easy, reliable and reproducible pathology FM benchmarking. It inherently supports a diverse range of public datasets, models, and a variety of metrics, while also offering flexibility for incorporating custom ones. All results in table 3 can be reproduced[3]. We are currently working on adding support for slide-level benchmark datasets together with segmentation tasks and other oncology-relevant modalities such as radiology.

## Acknowledgments

We thank Edwin D. de Jong, Iulia Lungu, Mikhail Karasikov, Axel Lagré, Joost van Doorn, Fei Tang and everyone in `kaiko.ai` for their support and fruitful discussions.

---

2. downsampled from 40× (0.25 µm/px)

3. https://kaiko-ai.github.io/eva/latest/user-guide/advanced/replicate_evaluations/

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
