# OpenReview forum: "eva: Evaluation framework for pathology foundation models"
_MIDL.io/2024/Short_Papers — MIDL 2024 Short Papers_

### Official Review · Reviewer_Cgnp · 2024-04-16

**Confidence:** 5
**Final Rating:** 5

**Review:**

The strengths of the paper revolve around the introduction of an evaluation framework specifically designed for pathology FM benchmarking. The framework's adaptability is highlighted by its inherent support for a diverse range of public datasets, models, and metrics, along with the flexibility to incorporate custom parameters. Furthermore, the results can be reproduced. These attributes position the paper well for acceptance, as it contributes a robust tool to the field and promises further advancements in the study of pathology FM.

---

### Decision · Program_Chairs · 2024-04-26

Accept